# Evaluating Pre-Interventional Administration of a Liver-Specific Contrast Agent During MRI-Guided Thermal Ablation of Malignant Liver Lesions

**DOI:** 10.3390/cancers17081264

**Published:** 2025-04-09

**Authors:** Antonia Ashkar, Jens Kübler, Konstantin Nikolaou, Rüdiger Hoffmann, Moritz T. Winkelmann

**Affiliations:** 1Department of Diagnostic and Interventional Radiology, University Hospital of Tuebingen, 72076 Tuebingen, Germanykonstantin.nikolaou@med.uni-tuebingen.de (K.N.); ruediger.hoffmann@med.uni-tuebingen.de (R.H.); moritz.winkelmann@med.uni-tuebingen.de (M.T.W.); 2Cluster of Excellence iFIT (EXC2180), Eberhard-Karls University, 72076 Tuebingen, Germany

**Keywords:** MRI, liver malignancies, thermal ablation, ablation zone, therapy monitoring, liver-specific contrast agent

## Abstract

Liver cancer and metastatic tumors can be treated with minimally invasive thermoablation procedures like microwave and radiofrequency ablation. Magnetic resonance imaging (MRI) helps guide these procedures by providing clear images of the tumor and surrounding tissue. In some cases, a hepatocyte-specific contrast agent is injected before the procedure to improve tumor visibility. However, this contrast agent may also make it harder to visualize the treated area after ablation, which is crucial for assessing treatment success. This study analyzed data from 57 patients (60 lesions), comparing those who received contrast agent before ablation (Group 1) to those who did not (Group 2). Results showed that contrast administration significantly improved lesion visibility but impaired the differentiation of the ablation zone. The findings will help determine whether using a contrast agent before ablation is beneficial or if it compromises treatment assessment. This research may guide future clinical decisions and improve MRI-based tumor treatment strategies.

## 1. Introduction

Percutaneous thermoablation techniques, such as microwave ablation (MWA) and radiofrequency ablation (RFA), are well-established, minimally invasive local treatment options for primary and secondary liver malignancies, especially for patients that are ineligible for surgery due to impaired liver function or advanced comorbidities [1,2]. The visualization of malignant liver lesion and the antenna artifact is crucial for the success of the ablation therapy, as is the ability to delineate the ablation zone from untreated tissue both during and after the procedure. In addition to widely used guidance modalities, such as computed tomography (CT) and ultrasound (US), ablation therapy can also be monitored using magnetic resonance imaging (MRI). The recent systematic review and network meta-analysis by Li et al., including 2349 patients, compared ultrasound-, CT-, and MRI-guided thermal ablation for hepatocellular carcinoma. MRI-guided ablation showed the highest probability of reducing local tumor recurrence (surface under the cumulative ranking curve [SUCRA] = 96.8) and improving primary technique effectiveness (SUCRA = 89.6), compared with CT (SUCRA = 35.8 and 34.6, respectively) and US (SUCRA = 17.4 and 14.5, respectively). SUCRA values estimate the probability of an intervention being the most effective or safest among all comparators. Although these differences were not statistically significant, MRI guidance was associated with a lower risk of major complications compared to CT (RR = 0.27; 95% CI: 0.13–0.59) and US (RR = 0.41; 95% CI: 0.10–1.74). Taken together, these findings suggest that MRI-guided ablation may offer favorable outcomes and a superior safety profile [3]. It also offers several advantages such as the unrestricted choice of angulation plane, near real-time fluoroscopy, increased soft tissue contrast, and the absence of ionizing radiation for both patient and radiologist [4,5,6]. Another study suggests that MRI-guided radiofrequency ablation (RFA) may be more effective than CT-guided ablation for treating hepatocellular carcinoma, potentially reducing the number of treatment sessions required [7]. Despite the availability of the less expensive alternatives such as CT and US, MRI, therefore, serves as a valuable imaging modality for percutaneous ablation therapy in selected cases [8,9].

Especially in liver ablation, high soft tissue contrast is beneficial [10]. Due to the above-mentioned advantages of MR imaging, it is often feasible to perform thermal ablation without pre-interventional contrast agent administration [11]. A study by Liu et al. showed the successful use of a microwave system in a 3.0 T MRI scanner [12], highlighting the importance of high-quality imaging to better assess lesions and ablation outcomes. However, for liver lesions that are difficult to delineate, hepatocyte-specific contrast agents such as Gd-EOB-DTPA can be administered, offering a long time window with increased visibility of the non-enhancing liver malignancies in comparison to the surrounding liver tissue [13,14]. Hammerstingl et al. showed that Gd-EOB-DTPA-enhanced MRI has a significantly higher rate of correctly detected liver lesions with small diameter below 1 cm compared to contrast enhanced biphasic CT [15]. Also, Liu et al. observed in a meta-analysis for HCC that lesions with a diameter of less than 2 cm can be detected significantly better with hepatocyte-specific contrast agent [16], suggesting a benefit during ablation procedures, especially when smaller liver lesions are targeted.

The application of hepatocyte-specific, gadolinium-containing contrast agents increases the T1 signal of the liver several minutes after administration by shortening the T1 relaxation time [17]. Similarly, the ablation zone has a typically hyperintense appearance in T1-weighted sequences, which can be used for therapy monitoring [18]. Consequently, the administration of hepatocyte-specific contrast agent may be beneficial for target lesion detection; however, assessment of the ablation result may be impaired.

To our knowledge, no previous study has directly evaluated the effect of pre-interventional administration of hepatocyte-specific contrast agent during real-time MRI-guided thermoablation on both target lesion visibility and ablation zone assessment. This study addresses this gap.

The purpose of this retrospective study was to evaluate the effect of pre-interventional administration of liver-specific contrast agent during MRI-guided thermal ablation on the visualization of the target tumor and the ablation zone.

## 2. Materials and Methods

### 2.1. Patient Population

Out of 358 patients that were treated during 2010 and 2020 with MRI-guided percutaneous thermoablation at our facility, 57 patients with a mean age of 61 years with 60 malignant liver lesions were retrospectively included in this study. Of those, 27 patients (22 male, 81%) with a total of 30 lesions (mean tumor size 17.5 ± 4.7 mm) in Group 1 received liver-specific MRI contrast agent Gd-EOB-DTPA (Primovist^®^, Bayer Schering Pharma, Berlin, Germany) prior to ablation for improvement of target lesions visibility. Another 30 patients (21 male, 70%) with a total of 30 lesions (mean tumor size 21.8 ± 3.2 mm) underwent MRI-guided thermoablation without pre-interventional administration of Gd-EOB-DTPA, forming control group 2. Figure 1 illustrates the study design flowchart, while Figure 2 outlines the timing of the ablation procedures in each group (Figure 1 and Figure 2). The entities of treated liver lesions were primary liver malignancies (HCC *n* = 23; 38%) and secondary liver malignancies: metastases of colorectal carcinoma (*n* = 20; 33%), metastases of malignant melanoma (*n* = 10; 17%) and other hepatic metastases (uveal melanoma *n* = 5; 8%, breast cancer *n* = 1; 2%, and renal cell carcinoma *n* = 1; 2%). In this study, most HCC patients (20 out of 23) had underlying liver cirrhosis due to hepatitis B or C infection, non-alcoholic steatohepatitis (NASH), or alcohol- and diet-related factors. At the time of the procedure, 14 of these 20 patients (70%) were classified as Child A stage.

Every treatment with local thermoablation was indicated by the institution’s tumor board. This retrospective study was approved by the Institutional Review Board of the University of Tübingen (approval code: 439/2023BO2).

### 2.2. Percutaneous Ablation Procedure

Thermoablation was conducted on 60 lesions, MWA for 31 lesions and RFA for 29 lesions.

RFA ablations were conducted using a bipolar radiofrequency system (Olympus Celon, Hamburg, Germany) with a 10 cm and 15 G applicator (group 1: *n* = 17, group 2: *n* = 12)). Depending on the size and number of lesions, one or two radiofrequency probes were positioned simultaneously. Microwave ablations were conducted with a non-perfusion-cooled, MRI-compatible microwave ablation system (Medwaves AveCure™, Medwaves, San Diego, CA, USA, group 1: *n* = 7, group 2: *n* = 3) operating at a frequency between 902 and 928 MHz; the generator power is controlled by a pulsatile power outlet. A temperature sensor 4 cm proximal to the tip of the antenna enables a feedback system to adapt the generator frequency and power so that a preselected temperature of 120 °C is generated at the tip of the antenna in a “temperature-controlled-mode”. The 14 G (gauge) antenna is not additionally cooled and operates at a maximum power of 36 watts.

During the study, a high-power MWA system with a generator output of 150 W and a frequency of 2.45 GHz (ECO-100E2, Nanjing ECO Medical Instrument Co., Nanjing, China), equipped with a perfusion pump for cooling the applicator shaft, was utilized during the procedure (group 1: *n* = 6, group 2: *n* = 15). Ablations were performed using an MR-compatible, 14-G microwave applicator (ECO-100AI13C, Nanjing ECO Medical Instrument Co., China) with a length of 15 cm. In all cases, the microwave generator was placed outside the scanner room during the procedures, with a coaxial cable connecting it to the MR-compatible antenna.

Ablation procedures were performed using two closed-bore 1.5 T MRI systems with a 70 cm bore diameter (Siemens Magnetom ESPREE and AERA, Siemens Healthineers, Erlangen, Germany), both equipped with a 32-channel body-phased-array coil.

An RF-shielded liquid crystal display monitor was positioned adjacent to the scanner’s bore, allowing real-time monitoring throughout the procedure, while the interventionalist remained positioned near the patient lying in the scanner.

### 2.3. MR-Guided Interventional Imaging

Pre-interventional planning imaging is acquired for visualization of the anatomical structures and the target tumor. The following unenhanced planning sequences were acquired: coronal T2 HASTE (half acquisition single shot turbo spin echo; TE 94 ms, TR 1100 ms, FA 160°, ST 3 mm, BW 488 Hz/pixel, FOV 380 × 380 mm^2^, matrix 320 × 320 mm^2^) and a fat-saturated (FS) T2-weighted STIR sequence (short time inversion recovery, TE 81 ms, TR 1400 ms, FA  160°, TI 180 ms, BW 449 Hz/pixel, ST 6 mm, FOV 340 × 340 mm^2^, matrix  384 × 384 mm^2^). If target lesion visualization was inadequate in the unenhanced planning sequences at the beginning of the interventional procedure (in group 1), hepatocyte-specific contrast agent (Gd-EOB-DTPA) was administered intravenously with a dose of 0.025 mmol/kg body weight. A fat-saturated unenhanced 3D T1-weighted VIBE (volumetric interpolated breath-hold imaging; TE 1.4 ms, TR 3.5 ms, FA 10°, ST 3 mm, BW 400 Hz/pixel, FOV 340 × 340 mm^2^, matrix 256 × 256 mm^2^) sequence was repetitively performed during the intervention. A MR-fluoroscopic sequence (BEAT-Multislice; TE 3.2 ms, TR 464 ms, FA 20°, BW 500 Hz/pixel, ST 8 mm, matrix 128 × 128 mm^2^) was acquired for near-real-time tracking of the applicator in three imaging orientations during tumor targeting. Thermoablation was started after confirmation of the correct applicator position. Therapy monitoring was also conducted with the unenhanced 3D T1-weighted VIBE. Ablation was repeated if the T1 hyperintense ablation zone showed insufficient coverage of the target tumor including an ablation margin of >5 mm. The applicator was withdrawn under coagulation if the ablation zone was considered adequate. Control imaging including a contrast-enhanced 3D T1-weighted Dixon VIBE after intravenous injection of Gadobutrol 0.1 mmol/kg body weight (Gadovist, Bayer HealthCare, Leverkusen, Germany) was then performed in both groups to confirm technical success and rule out complications.

### 2.4. Statistical Analysis

Images were analyzed using Centricity PACS (RA1000, GE HealthCare Technologies, Chicago, IL, USA). The signal intensity (SI) of the target lesion, the adjacent liver parenchyma, and the peripheral ablation zone was measured with ROIs (region of interests) in unenhanced sequences before applicator positioning, during unenhanced therapy monitoring and at Gadobutrol-enhanced control imaging. Care was taken to place the ROI in homogeneous liver tissue and to avoid vessels. Each position was measured three times, and a mean SI was determined. Contrast-to-noise ratio (CNR) was calculated with the following formula, whereby the standard deviation was measured placing a ROI in homogeneous liver parenchyma close to the lesion respectively the ablation zone:CNR = (SI_1_ − SI_2_)/SD

The CNR of the target lesion prior to applicator positioning, the ablation zone during unenhanced therapy monitoring, and the ablation zone during contrast-enhanced control imaging were determined. Comparison of the CNR values for these three structures between group 1 and group 2 was conducted using the non-parametric Wilcoxon test with a significance threshold of *p* < 0.05. Data analysis was performed using JMP (Version 14.2.0, SAS Institute Inc., Cary, NC, USA) and SPSS 22.0 (IBM Corp., Armonk, NY, USA). Continuous variables are presented as mean ± standard deviation (SD). Non-normally distributed data are represented by the median value and the interquartile range.

## 3. Results

During planning imaging, the contrast-to-noise ratio (CNR) of the target lesions in group 1 (contrast agent administered before ablation) increased significantly after the administration of Gd-EOB-DTPA, with CNR values rising from a median of 2.5 to 6.3 (*p* = 0.0006); an example of clinical application is illustrated in Figure 3.

After the administration of Gd-EOB-DTPA in group 1, no significant difference in the CNR was observed in comparison with group 2 (no contrast agent administered before ablation) for the target lesions, with CNR values of 6.3 in group 1 versus 6.15 in group 2 (*p* = 0.684).

During therapy monitoring with unenhanced T1 3D VIBE, the CNR of the ablation zone was significantly higher in group 2, with a median CNR of 7.9 compared to 2.1 in group 1 (*p* < 0.001).

After the administration of Gadobutrol for control imaging after ablation, the CNR of the ablation zone was significantly higher in group 2, with a median value of 7.7 compared to 2.0 in group 1 (*p* < 0.001) (Figure 4).

Table 1 summarizes the CNR results for both target lesions and ablation zones, along with the corresponding *p*-values. The distribution of CNR values across different groups is illustrated in Figure 5.

## 4. Discussion

The aim of this study was to evaluate the contrast-to-noise ratio of liver lesions and ablation zones in patients with and without the administration of hepatocyte-specific contrast agent (Gd-EOB-DTPA) at different timepoints during MRI-guided thermoablation. Our results show a clear improvement in the delineation of the liver lesions in group 1 after the administration of the hepatocyte-specific contrast agent, which could then be visualized similarly to the lesion in group 2, in which no contrast agent was administered. Nevertheless, there was a significantly improved delineation of the hyperintense ablation zone in the group in which no contrast agent was administered prior to ablation. This improved demarcation was visible both during the monitoring of the therapy and in the final control images compared to the group that had received hepatocyte-specific contrast agent prior to ablation.

Liver-specific contrast agents result in a shortening of the T1 relaxation time and, therefore, cause an enhanced T1 signal, which improves the delineation of small focal liver lesions [19]. It is taken up by the hepatocytes and excreted over the biliary system. Unlike extracellular contrast agent, which has only a short time window, liver-specific contrast agent shows the lesion clearly during the so-called hepatobiliary phase for up to several hours [20], providing the interventionalist sufficient time for an MRI-guided microwave ablation [21]. However, as demonstrated in this study, it may result in reduced differentiation of the T1 hyperintense ablation zone from the surrounding untreated liver parenchyma. This limitation can hinder the immediate assessment of the therapeutic effect and potentially increase the risk of a higher local recurrence rate. The cases in which liver-specific contrast agent was administered in group 1 were more challenging to detect compared to the lesions in the non-contrast group 2, which may indicate a potential pre-selection bias. An increase in contrast between the lesion and surrounding liver parenchyma due to the elevated T1 signal after the administration of the liver-specific contrast agent in group 1 might result in an approximation and a more uniform appearance and, therefore, compensating the previously existing differences in the delimitation of the lesions, possibly explaining that there was no statistically significant difference between both groups in the imaging directly prior to ablation.

When discussing the use of pre-interventional contrast agents, it should also be borne in mind that liver-specific contrast agent (Gd-EOB-DTPA) itself is more expensive then extracellular contrast agents such as Gadovist^®^ [22] and the additional sequences during and after the administration of liver-specific contrast agent prolong the overall duration of the ablation procedure, which has a further negative impact on cost-effectiveness, particularly in MRI-supported ablation therapy where the time required is already high [23].

In their study, Fischbach et al. showed the process of administering liver-specific contrast agent (Gd-EOB-DTPA) to each patient 20 min prior to the MR-guided intervention and, therefore, acquiring all imaging in the hepatobiliary phase, resulting in a procedure time of 30–60 min [24]. However patients with impaired liver function, e.g., with a bilirubin value > 3 mg/dL, show a delayed and reduced uptake of the liver-specific contrast agent [13]. As patients who require ablation therapy often develop liver malignancies on the basis of an underlying fibrocirrhotic disease, the positive detection effect of the liver-specific contrast agent is limited [25]. In addition, there can be contradictions for the administration of contrast agents, such as rare but possibly severe allergic reaction, advanced renal insufficiency (GFR < 15) [26,27], or nephrogenic systemic fibrosis (NSF) [28].

It would be valuable to evaluate and compare the costs against the potential benefits, e.g., fewer treatment sessions due to better delineation of the lesions. In situations where small lesions are not clearly visible on T1 VIBE imaging, an alternative approach without contrast administration but using inversion recovery (IR) imaging could be considered. By adjusting the individual inversion time (TI), a greater contrast between the liver parenchyma and malignant liver lesions can be achieved [29]. A study by Guo et al. demonstrated that after irreversible electroporation (IRE), IR-prepared sequences can be used to differentiate irreversibly electroporated liver tissue (central ablation zones) from reversibly electroporated peripheral tissue (penumbra) and, therefore, provide more accurate ablation zone measurements compared to conventional T1-weighted MR imaging [30].

The results of this study are limited by several factors, such as the retrospective study design and the relatively small study population, whereas the limiting factor is the low number of cases with pre-interventional administration of contrast agent at our institute. Another limitation is the different initial conditions of both groups, as the lesions in group 1 were not naturally well visible and, therefore, a liver-specific contrast agent was administered prior to the intervention. The study spans a 10-year period, during which technical advancements in our department led to the implementation of different thermal ablation systems (radiofrequency and microwave ablation) as well as the use of two different MRI scanners. Both MRI systems were from the same vendor and operated at 1.5 Tesla, which helps to ensure consistency in image quality despite the change in hardware. Nonetheless, this technical heterogeneity may have introduced a degree of variability and represents a limitation of the study. Although underlying cirrhosis, particularly in advanced stages such as Child-Pugh B or C, can alter the distribution and dynamics of contrast agents in the fibrotic liver parenchyma [31,32], only a minority of patients in our cohort had cirrhosis, and the majority of these were classified as Child-Pugh A. Therefore, while subtle effects on contrast behavior cannot be ruled out, we consider the overall impact on the results to be limited.

Future studies with larger patient cohorts should aim for a more distinct randomization between MRI-guided thermoablation with and without intraprocedural Gd-EOB-DTPA, ideally including analyses of recurrence and survival rates where applicable.

## 5. Conclusions

The pre-interventional administration of a hepatocyte-specific contrast agent (Gd-EOB-DTPA) improves the visualization of poorly demarcated liver lesions but significantly reduces the contrast of the ablation zone during intra- and post-procedural imaging. Therefore, its use should be reserved for cases in which the target lesion cannot be sufficiently delineated without contrast.

## Figures and Tables

**Figure 1 cancers-17-01264-f001:**
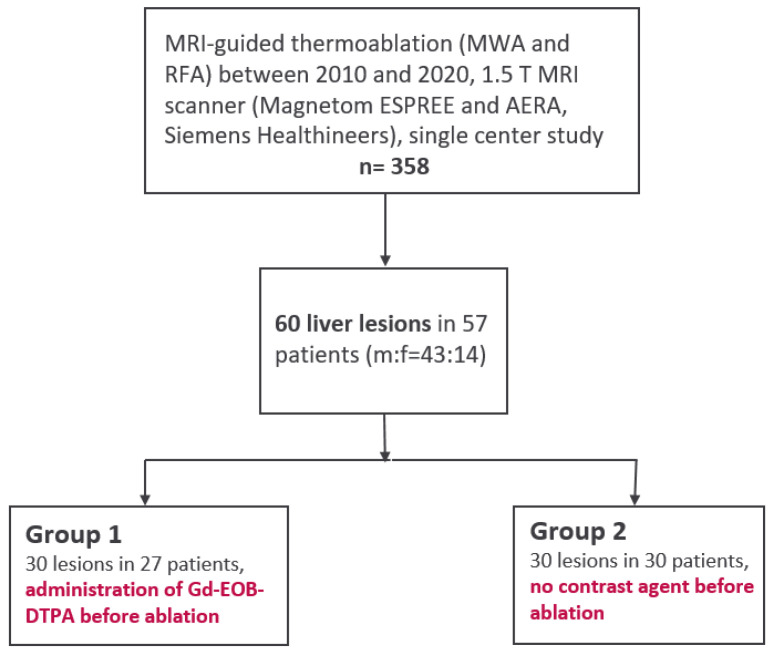
Flowchart of the study design.

**Figure 2 cancers-17-01264-f002:**
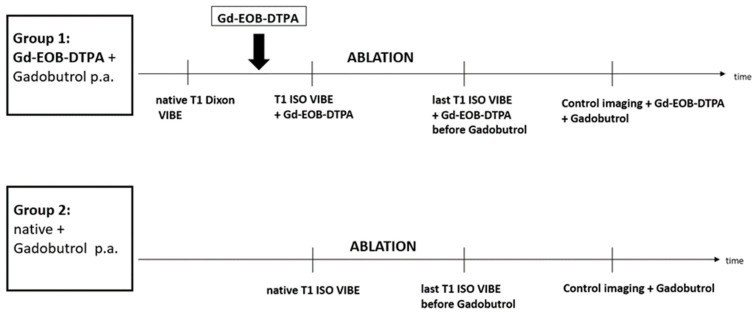
Timeline of interventional procedure: group 1 with the administration of Gd-EOB-DTPA before ablation and group 2 the native control (p.a. post ablation).

**Figure 3 cancers-17-01264-f003:**
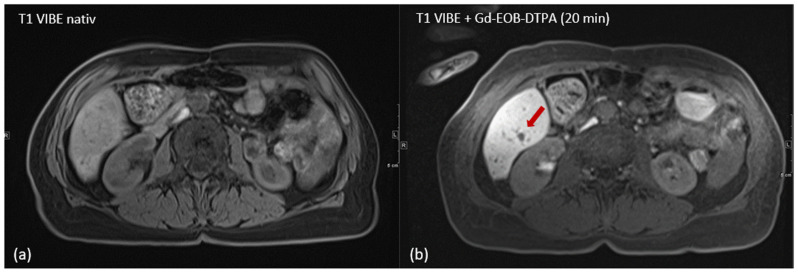
Hepatic metastasis in a 63-year-old female patient with colon cancer in liver segment VI; the unenhanced T1 VIBE sequence shows a very poor demarcation of the lesion (**a**) and significantly improved delineation of the hypointense liver lesion (arrow) in the hepatobiliary phase 20 min after administration of hepatocyte-specific contrast agent (Gd-EOB-DTPA) with T1 hyperintense liver parenchyma (**b**).

**Figure 4 cancers-17-01264-f004:**
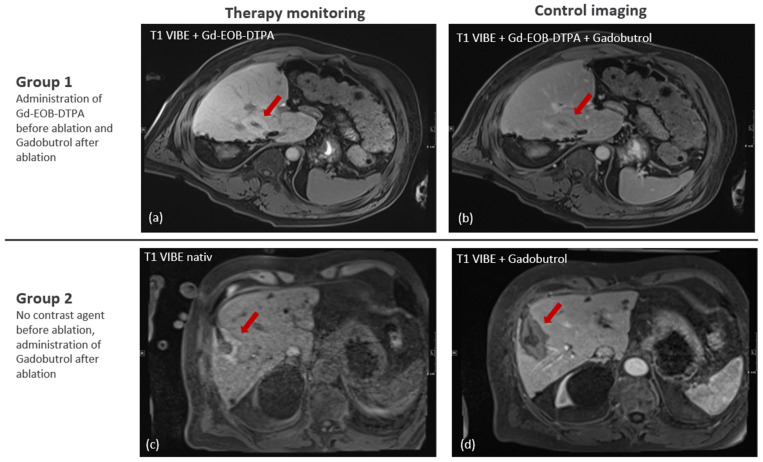
(**a**,**b**) 52-year-old male patient of group 1 with a HCC lesion in liver segment VIII: microwave ablation after administration of Gd-EOB-DTPA with a slightly hyperintense ablation zone (**a**) and nearly isointense ablation zone after also administering Gadobutrol during control imaging (**b**) and, therefore, difficult demarcation of the ablation. (**c**,**d**) Hepatic metastasis of an 80 year old male patient with a malignant melanoma in liver segment V/VIII after microwave ablation in an unenhanced T1 VIBE sequence with an hyperintense ablation zone that covers the hypointense lesion completely (**c**) and the same ablation zone hypointense in a T1 VIBE sequence after administration of contrast agent Gadobutrol (**d**) with improved delineation of the liver lesion an the ablation zone in group 2 compared to group 1.

**Figure 5 cancers-17-01264-f005:**
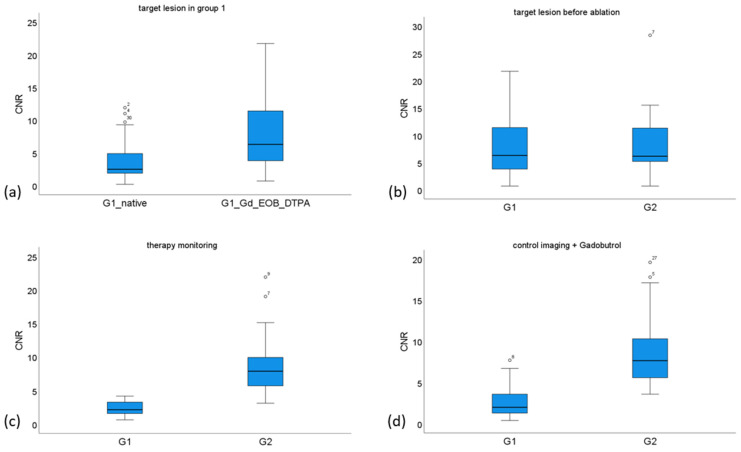
Box plots with the median and the 25th and 75th percentile of the CNR of the target lesion in group 1 (G1) before (G1_native) and after administering Gd-EOB-DTPA (G1_Gd-EOB-DTPA) (**a**), the CNR of the target lesion of group 1 and 2 (G2) (**b**), the CNR of the ablation zone (AZ) during therapy monitoring of group 1 and 2 (**c**), and the CNR during the final control imaging after administering Gadobutrol (**d**).

**Table 1 cancers-17-01264-t001:** Median value of the contrast-to-noise-ratio (CNR) of the target lesion, the ablation zone during therapy monitoring and during control imaging of group 1 and 2, and the corresponding *p*-value (* = significant result).

	Group 1 Gd-EOB-DTPA Before Ablation Gadobutrol After Ablation		Group 2 Gadobutrol After Ablation	*p*-Value (*p* < 0.05 *)
CNR target lesion	native 2.5	+ Gd-EOB-DTPA 6.3		*p* = 0.0006 *
CNR of ablation zone during therapy monitoring	2.14		7.85	*p* < 0.0001 *
CNR of ablation zone during control imaging	2.0		7.65	*p* < 0.0001 *

## Data Availability

The raw data supporting the conclusions of this article will be made available by the authors on request.

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
