# Peer review of "Evaluating Pre-Interventional Administration of a Liver-Specific Contrast Agent During MRI-Guided Thermal Ablation of Malignant Liver Lesions"

_cancers, 2025, doi:10.3390/cancers17081264_

Round 1
Reviewer 1 Report
Comments and Suggestions for Authors
Evaluating pre-interventional administration of a liver-specific 2 contrast agent during MRI-guided thermal ablation of malignant liver lesions. My comments are as follows:
- What were the etiologies of HCC in this study and what fraction of patients were cirrhosis vs no-cirrhosis?
- The authors have rightly mentioned lack of randomization is an important limitation of the study. Could the etiology of underlying HCC, and presence of absence of cirrhosis influence the results?
- Were there any differences based on the size of tumor and prior therapy received? Did any patients receive anti-cancer drugs prior to intervention
- Check for spellings and grammar eg. The following statement in unclear "with improved delineation of the liver le-sion an the ablation zone in group 2 compared to group 1.”
- It is unclear what the authors mean to depict in the table above figure 5. y, n, 2,5, 2,14. What do each of these figures mean?
- Clarify G1_native
Needs improvement
Author Response
Dear Reviewer 1,
Thank you very much for taking the time to review our paper and for the helpful comments improving the quality of the paper. All alterations were highlighted yellow in the manuscript.
- What were the etiologies of HCC in this study and what fraction of patients were cirrhosis vs no-cirrhosis?
Thank you for this aspect, among the 57 patients, 23 had HCC, with 20 of them developing it due to cirrhosis caused by ethyl toxicity or dietary factors, hepatitis B or C infection, or NASH. Of these 20 patients, 14 were classified as Child A stage. This information has been added to page 4.
- The authors have rightly mentioned lack of randomization is an important limitation of the study. Could the etiology of underlying HCC, and presence of absence of cirrhosis influence the results?
Thank you for this important point. In our cohort, only approximately one third of patients had underlying cirrhosis, with the majority of these (70%) classified as Child-Pugh A. Given the relatively low proportion of advanced liver disease and the fact that no systematic differences in lesion conspicuity or ablation zone delineation were observed between cirrhotic and non-cirrhotic patients in our dataset, we consider it unlikely that cirrhosis significantly influenced the results. However, we acknowledge that advanced cirrhosis (Child-Pugh B or C) can affect contrast dynamics due to altered hepatic perfusion. This potential confounder is addressed as a limitation in the revised manuscript (p. 10).
Two pictures (shown in the attached documt) give an example of a cirrhotic liver (a) Child Pugh B and a non-cirrhotic liver (b), both patients from group 1 with rather similar CNR levels (CNR 0,97 (a) vs. CNR 1,37 (b)) and a subjective similar delineation of the ablation zone against the adjacent liver parenchyma.
- Were there any differences based on the size of tumor and prior therapy received? Did any patients receive anti-cancer drugs prior to intervention
Thank you for pointing this out, the mean size of the treated tumor has been added (group 1: 17,5 ± 4,7 mm, group 2: 21,8 ± 3,2 mm, p. 3 and p.4). According to established guidelines, microwave ablation (MWA) and radiofrequency ablation (RFA) are typically recommended for treating liver tumors up to 3 cm in diameter. HCCs with a size >10 mm should be treated. Consequently, the applicable treatment range for these modalities spans approximately 1 to 3 cm. In our study, the mean tumor size prior to intervention falls squarely within this optimal treatment range.
Some of the patients had additional anti-cancer drugs as part of their therapy, some of which has been ongoing for many years. As the study population of 57 patients with several different tumor entities yet very similar diagnostic characteristics in MRI imaging is pretty heterogeneous, a subdivision into individual drug therapies would have resulted in very small group sizes and a separate evaluation was therefore not carried out.
- Check for spellings and grammar eg. The following statement in unclear "with improved delineation of the liver le-sion an the ablation zone in group 2 compared to group 1.”
Thank you for bringing this to our attention. A comprehensive spelling and grammar review has been conducted, although the correction of minor spelling mistakes was not highlighted in the manuscript.
- It is unclear what the authors mean to depict in the table above figure 5. y, n, 2,5, 2,14. What do each of these figures mean?
Thank you very much for pointing this out. The numbers in the table are the median value of the CNR for the different points of observation (target lesion, ablation zone during therapy monitoring and control imaging) in group 1 and group 2, which has been added to the table caption. Y (yes) and n (no) has been removed of the table and table caption, hoping that this will help to clarify the statement of the table
- Clarify G1_native
G1_native describes the time point in group 1 before the contrast agent has been administered to visualize the target lesion, hence in native resp. unenhanced imaging. A specific description has been added to the caption of Figure 5, hopefully clarifying this aspect.

Reviewer 2 Report
Comments and Suggestions for Authors
General comment:
This work deals with the pre-interventional use of MRI contrast agents to drive and guide the MW ablation of liver tumors.
The study was carried out on 60 lesions, divided in two groups. The results highlight that the use of the contrast agent improved the lesion visibility but impaired the differentiation fo the ablation zone.
Specific comments throughout the paper:
The frequencies for MWA and RFA are not reported.
The introduction section is too qualitative and not quantitative, poorly relying on the use and analysis fo figures of merit to highlight the literature lacks. Furthermore, the introduction is too short and poorly focused resulting in a scarce novelty. It must be improved.
The authors are not reporting if the MRI scanner and RF coils are all the same for all subjects. The variability must be commented and included.
Furthermore, why using two very different thermal ablation systems is not faed and discussed. The variability in the electromagnetic field, specific absorption rate and the heating patterns between the two treatment modalities is large.
The author are not commenting on the selection of the concentration and time of the contrast agent. Also, more details and ref. for the scanning and other methods must be provided.
Using only CNR as a figure of merit is not enough to validate and support the study hypothesis. Therefore, the results demands for further support.
Minor edits:
The paper has several editing and formatting issues. Please revise accordingly to the instructions.
The paper organization and section headings and numbering can be revised.
Author Response
Dear Reviewer 2,
Thank you very much for taking the time to review our paper and for the helpful comments improving the quality of the paper. All alterations were highlighted light blue in the manuscript.
Specific comments throughout the paper:
- The frequencies for MWA and RFA are not reported.
The frequencies of MWA (31 lesions) and RFA (29 lesions) can be found on page 4.
- The introduction section is too qualitative and not quantitative, poorly relying on the use and analysis fo figures of merit to highlight the literature lacks. Furthermore, the introduction is too short and poorly focused resulting in a scarce novelty. It must be improved.
We appreciate your constructive feedback. We have substantially revised the introduction to provide a more focused and quantitatively underpinned background. In particular, we have added important results from a recent network meta-analysis of 2349 patients (Li et al.), including numerical data such as SUCRA rankings, relative risks, and confidence intervals. This allows a more meaningful comparison between imaging modalities and highlights the potential advantages of MRI guidance in ablation therapy. Furthermore, we have emphasized the novelty of our study by identifying the gap in the literature regarding the effect of hepatocyte-specific contrast agents during real-time MR-guided thermoablation. We hope that these revisions sufficiently address your concerns.
- The authors are not reporting if the MRI scanner and RF coils are all the same for all subjects. The variability must be commented and included.
“Ablation procedures were conducted in two closed bore 1,5 T MRI systems with a bore diameter of 70 cm (Siemens Magnetom ESPREE and AERA, Siemens Healthineers, Erlan-
gen, Germany) equipped with a 32-channel body-phased-array coil.” (p. 5)
Thank you very much for pointing this out. The same coil was used for all procedures. In the period from 2010 to 2020 our department purchased a new MRI scanner, so the procedures were performed first on a Siemens Espree MRI (1.5 T) and later on a Siemens AERA (also 1.5 T), both machines from the same manufacturer and with the same magnetic field strength, so differences are of course possible and are also a limitation but do not necessarily have to be significant. We now addressed this potential confounder as a limitation in the revised manuscript.
- Furthermore, why using two very different thermal ablation systems is not faed and discussed. The variability in the electromagnetic field, specific absorption rate and the heating patterns between the two treatment modalities is large.
Thank you for bringing up this aspect. Our study covers a long time period (2010 – 2020) in order to ensure a large study population. Technological progress and investments have allowed our department to purchase a new ablation device during that time period. We have added this aspect to the limitations of our study (page 9/10).
- The author are not commenting on the selection of the concentration and time of the contrast agent. Also, more details and ref. for the scanning and other methods must be provided.
We described the intravenously administered concentration of Gd-EOB-DTPA of 0,025 mmol/kg body weight, we added a description of the general time of administration of the Gd-EOB-DTPA if target lesion visualization was inadequate in the unenhanced planning sequences at the beginning of the interventional procedure (p.5). In subsection 2.2 (Percutaneous ablation procedure) and 2.3 (MR guided interventional imaging) we give detailed information about the scanning protocol, ablation devices and procedural set-up in the scanner room.
- Using only CNR as a figure of merit is not enough to validate and support the study hypothesis. Therefore, the results demands for further support.
Thank you for your valuable comment. We agree that CNR alone may not comprehensively reflect all aspects of image quality. Nevertheless, contrast-to-noise ratio (CNR) is a well-established and objective metric for assessing lesion conspicuity, particularly in liver imaging, and has been used in comparable studies (e.g., Chung et al., 2006 https://doi.org/10.1002/jmri.20557 ; Goenka et al., 2016 doi.org/10.1148/radiol.2016151621). Given the limited sample size and retrospective nature of our study, we aimed to focus on a robust and reproducible parameter that is feasible in a clinical setting. More advanced analyses, such as texture-based or reader-based assessments, were considered beyond the scope of this study due to statistical limitations and heterogeneity. However, we fully agree that future prospective studies with larger and more homogeneous populations should incorporate complementary image quality metrics to further validate and support the findings.
Minor edits:
- The paper has several editing and formatting issues. Please revise accordingly to the instructions. The paper organization and section headings and numbering can be revised.
Thanks for pointing out these issues, the editing and formatting has been revised according to the instructions for authors, and the headings and numbering have been revised.

Reviewer 3 Report
Comments and Suggestions for Authors
This is the clinical study of the liver specific contrast agent used for targeting liver and observations by MRI imaging. The study is good and does show the enrichment of the molecule in liver but there are other side distributions as well. Also, I am surprised that since liver problems could be easily detected by ultrasound or ct-scan, what new technology does this study provides. A clear mention of the novelty of this method is required for the manuscript.
Also the data is more qualitative pictures than the quantification. The statistical distribution of the data is also missing from the paper that needs to be corrected.
Author Response
Dear Reviewer 3,
Thank you very much for taking the time to review our paper and for the helpful comments improving the quality of the paper. All alterations were highlighted pink in the manuscript.
This is the clinical study of the liver specific contrast agent used for targeting liver and observations by MRI imaging. The study is good and does show the enrichment of the molecule in liver but there are other side distributions as well.
- Also, I am surprised that since liver problems could be easily detected by ultrasound or ct-scan, what new technology does this study provides. A clear mention of the novelty of this method is required for the manuscript.
Thank you for your comment. We have substantially revised the introduction to provide a more focused and quantitatively underpinned background. While ultrasound and CT are widely used for liver interventions, MRI offers distinct advantages such as superior soft tissue contrast, multiplanar capabilities, and the absence of ionizing radiation. Although MRI-guided liver ablations have been described in the literature, the number of studies remains limited compared to CT- and US-guided procedures, and standardized protocols are lacking.
The novelty of our study lies in the evaluation of hepatocyte-specific contrast agent administration prior to MRI-guided thermal ablation—an area that has not been systematically studied to date. Specifically, we investigate its impact on lesion conspicuity and ablation zone assessment, which are critical for procedural success and treatment monitoring. This contributes to the ongoing development of optimized protocols for MR-guided liver interventions, particularly in centers with growing expertise in this technique.
- Also the data is more qualitative pictures than the quantification. The statistical distribution of the data is also missing from the paper that needs to be corrected.
We used signal intensity and contrast-to-noise ratio as common methods to validate our data. Thank you for pointing this out, information about the statistical distribution of the data has been added resp. described more precisely on page 7 at the subsection 2.4 Statistical analysis.

Round 2
Reviewer 1 Report
Comments and Suggestions for Authors
The revised manuscript is much improved. Few points
- The authors have used . / , interchangeably in the manuscript. In order for the clear meaning of units use . uniformly instead of ,
- The conclusion needs to be made more clear. This sentence lacks meaning- Although target lesion delineation was often improved, its use should be limited to selected cases.
Needs Improvement
Author Response
Dear Reviewer 1,
Thank you for taking the time once again to review the revision of our paper and for your helpful comments.
The revised manuscript is much improved. Few points
- The authors have used . / , interchangeably in the manuscript. In order for the clear meaning of units use . uniformly instead of ,
Thank you for pointing this out, all characters have been standardized and commas have been replaced by periods.
2. The conclusion needs to be made more clear. This sentence lacks meaning- Although target lesion delineation was often improved, its use should be limited to selected cases.
Thank you for this comment, we have revised the conclusion and rephrased it to clarify the message of our paper. Changes are again highlighted in yellow.
Reviewer 2 Report
Comments and Suggestions for Authors
I do not have any further comments
Author Response
Dear Reviewer 2,
Thank you for taking the time to review our paper and for the helpful comments that have improved the quality of our work.